# Observation and manipulation of quantum interference in a superconducting Kerr parametric oscillator

Daisuke Iyama[1,2,7], Takahiko Kamiya[1,2,7], Shiori Fujii[1,2,7], Hiroto Mukai [2,3], Yu Zhou[2], Toshiaki Nagase[1,2], Akiyoshi Tomonaga [2,3], Rui Wang [2,3], Jiao-Jiao Xue[2,4], Shohei Watabe [5], Sangil Kwon [3] ✉ & Jaw-Shen Tsai[2,3,6]

Quantum tunneling is the phenomenon that makes superconducting circuits "quantum". Recently, there has been a renewed interest in using quantum tunneling in phase space of a Kerr parametric oscillator as a resource for quantum information processing. Here, we report a direct observation of quantum interference induced by such tunneling and its dynamics in a planar superconducting circuit through Wigner tomography. We experimentally elucidate all essential properties of this quantum interference, such as mapping from Fock states to cat states, a temporal oscillation due to the pump detuning, as well as its characteristic Rabi oscillations and Ramsey fringes. Finally, we perform gate operations as manipulations of the observed quantum interference. Our findings lay the groundwork for further studies on quantum properties of superconducting Kerr parametric oscillators and their use in quantum information technologies.

The quantum tunneling of charge or flux degrees of freedom, once studied for academic interest[1,2], is now the basis of superconducting quantum technology[3–8]. Controlling quantum tunneling in phase space is also the key to the Hamiltonian engineering of a Kerr nonlinear resonator with a two-photon pump, commonly referred to as a Kerr parametric oscillator (KPO)[9,10]. In recent years, extensive research has been dedicated to exploring the properties of this exotic driven quantum system due to its promising applications in quantum computation[11–17], quantum annealing[18–25], and quantum error correction[26–29].

This potential usefulness relies on three fundamental properties of a KPO: (i) Schrödinger cat states are formed by quantum tunneling between two states confined by the Hamiltonian itself, yielding an interference pattern in Wigner tomography within a classically forbidden region[12,18,26,30–33]. (Throughout this work, we refer to this interference pattern in phase space as quantum interference.) (ii)

Conventional Fock-state encoding and cat-state encoding exhibit one-to-one mapping that preserves quantum coherence. (iii) Gate operations on the cat states are simple and intuitive, akin to those on a two-level system (TLS). These properties distinguish cat states in a KPO from other cat states generated by different methods, such as dynamic generation[34–36], interaction with a TLS[37,38], and dissipation engineering[39–41].

Although there have been numerous experimental works on a superconducting KPO[42–46], which is the system of our interest, and now it is being actively used for practical applications[47–50], only a few experimental works on its intrinsic quantum properties have been reported[51–54]. In particular, experimental observation of the quantum interference remains elusive. The first experimental attempt to observe such quantum interference was reported in Ref. 51. However, the state characterization method used in this work, transient power spectral density[33], requires the system to strongly couple to the

[1]Department of Physics, Graduate School of Science, Tokyo University of Science, Shinjuku-ku, Tokyo, Japan. [2]RIKEN Center for Quantum Computing (RQC), Wako-shi, Saitama, Japan. [3]Research Institute for Science and Technology, Tokyo University of Science, Shinjuku-ku, Tokyo, Japan. [4]Institute of Theoretical Physics, School of Physics, Xi'an Jiaotong University, Xi'an, People's Republic of China. [5]College of Engineering, Department of Computer Science and Engineering, Shibaura Institute of Technology, Koto-ku, Tokyo, Japan. [6]Graduate School of Science, Tokyo University of Science, Shinjuku-ku, Tokyo, Japan. [7]These authors contributed equally: Daisuke Iyama, Takahiko Kamiya, Shiori Fujii. ✉e-mail: kwon2866@gmail.com

environment. As a result, quantum coherence is significantly suppressed, making adiabatic cat state generation difficult[12,18,26,33,55,56]. Another approach, taken in Ref. 52, involves using a readout resonator for state characterization, thus preserving the quantum coherence of the KPO. Although this work demonstrated mapping from Fock states to cat states and single-qubit gate operations, these demonstrations relied on support from simulations, as the dispersive readout does not provide full information on the quantum state of the KPO during the activation of the pump.

In this work, we develop methods for complete quantum state characterization of a planar superconducting KPO. We report direct experimental observations of the quantum interference and demonstrations of mapping from Fock states to cat states via Wigner tomography (Lutterbach–Davidovich method[57]) without any supports from simulations or prior assumptions. We also investigate dynamics of this quantum interference by observing a temporal oscillation induced by

the pump detuning as well as Rabi oscillations and Ramsey fringes of cat states. Finally, we implement single-qubit gate operations. We characterize the mapping and gate operation by quantum process tomography (QPT).

## Results

### System characterization

Standard techniques for the state characterization of a superconducting TLS and quantum harmonic oscillator (QHO) are currently dispersive measurement and Wigner tomography, respectively. The problem of dispersive measurement is that the dispersive shift of an ancillary QHO, often called a readout resonator, may be very small for a system of our interest, whose self-Kerr coefficient is typically less than 1% of its transition frequency[6]. Therefore, we utilize an ancillary nonlinear system[58], a transmon (Fig. 1a), whose nonlinearity is orders of magnitude greater than that of the target KPO[38]. The coupling between

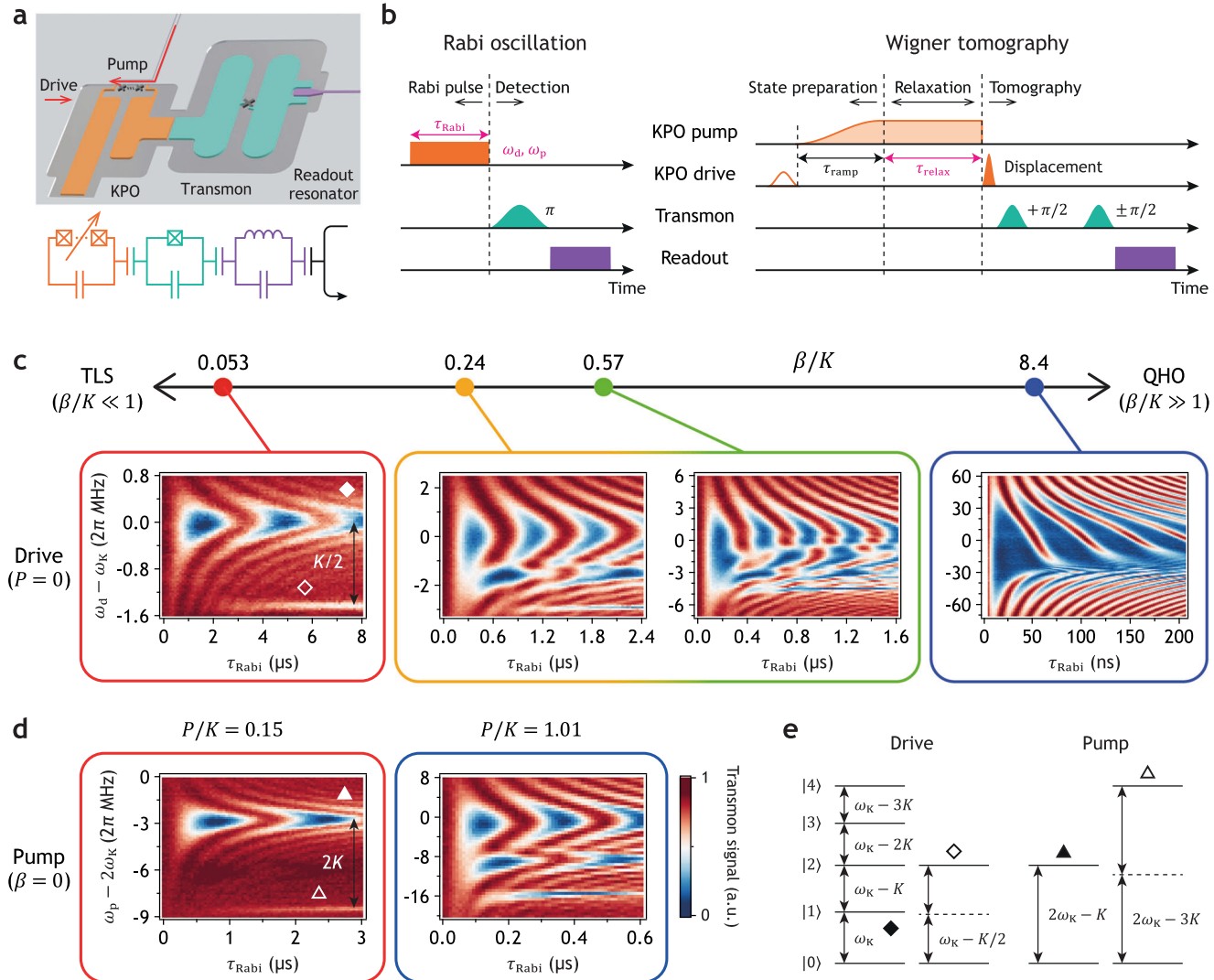

**Fig. 1 | System characterization. a** Rendered drawing of the chip and its circuit diagram. A cross symbol represents a Josephson junction. The KPO (orange) is composed of a series of 10 DC superconducting quantum interference devices (DC SQUIDs) with a shunting capacitor. This series of DC SQUIDs is indicated by two crosses with dots. The transmon (green) is capacitively coupled to the KPO. The readout resonator (purple) is a quarter-wavelength resonator. The system parameters can be found in Supplementary Table 1. **b** Pulse sequence for Rabi oscillations and Wigner tomography. The control parameters for the Rabi oscillation measurements are shown in magenta. "$\pi$" and "$\pi/2$" mean the transmon $\pi$- and

$\pi/2$-pulses, respectively. For the Wigner tomography, the first pulse in the KPO drive (empty orange pulse) is the Fock state preparation pulse. The Wigner tomography pulse sequence is used to obtain the data in Figs. 2 and 3. The pulse conditions are summarized in Supplementary Table 2. **c, d** Rabi oscillations in the $|0\rangle$ state population of the KPO driven by the drive (c) and the pump (d). TLS and QHO stand for two-level system and quantum harmonic oscillator, respectively. **e** Energy level diagram of the KPO. The correspondence between the transitions and the observed signals is indicated by diamonds and triangles.

the KPO and the transmon induces photon number splitting in the transmon spectrum (see Supplementary Fig. 3d). This allows us to detect the population of the Fock basis[59] and the number parity of the KPO using the pulse sequences shown in Fig. 1b. While the Wigner function represents the number parity as a function of phase-space coordinate[60], in a KPO, measuring the number parity with displacement pulses does not guarantee a reliable measurement of the Wigner function.

The challenge in performing Wigner tomography on a KPO arises from the distortion caused by non-commutativity between the Kerr nonlinear term and the single-photon drive term used for the displacement operation. (Hereafter, we refer to the single-photon drive term as the drive.) Due to the rapid evolution of the quantum state of a KPO during the displacement pulse, significant distortion occurs in the tomography results. To address this issue, we apply short and strong displacement pulses allowing for postprocessing to correct the tomography distortion (see Sec. 5A in Supplementary Information).

In a frame rotating with the frequency $\omega_p/2$, where $\omega_p$ is the frequency of a two-photon pump, the Hamiltonian of the KPO is given by (see Sec. 1 in Supplementary Information for the derivation)

$$\hat{\mathcal{H}}(t)/\hbar = \Delta(t)\,\hat{a}^\dagger\hat{a} - \frac{K}{2}\,\hat{a}^\dagger\hat{a}^\dagger\hat{a}\hat{a} + \frac{P(t)}{2}\left(\hat{a}^\dagger\hat{a}^\dagger + \hat{a}\hat{a}\right)$$
$$+ \beta(t)\left[\hat{a}^\dagger e^{-i(\Delta_d t + \phi_d)} + \hat{a}\,e^{+i(\Delta_d t + \phi_d)}\right]. \tag{1}$$

Here, $\hat{a}$ and $\hat{a}^\dagger$ are the ladder operators for the KPO; $\Delta (\equiv \omega_K - \omega_p/2)$ is the KPO-pump detuning, where $\omega_K$ is the transition frequency between the $|0\rangle$ and $|1\rangle$ states; $K$ is the self-Kerr coefficient; $P$ is the amplitude of the pump; $\beta$ is the amplitude of the drive; $\Delta_d (\equiv \omega_d - \omega_p/2)$ is the drive-pump detuning, where $\omega_d$ is the frequency of the drive; and $\phi_d$ is the phase of the drive relative to the half of the pump phase.

The validity of this model is confirmed by measuring Rabi oscillations in the population of the KPO $|0\rangle$ state (Fig. 1c). The dynamics of the KPO are primarily governed by two parameters: $\beta/K$ and $P/K$. In the limit of $\beta/K \ll 1$ (with $P/K = 0$ for simplicity), the dynamics of the KPO closely resemble those of a TLS (the leftmost plot in Fig. 1c). The Rabi oscillation denoted by the filled diamond is induced by the transition between $|0\rangle$ and $|1\rangle$ states described in Fig. 1e. The weak signal denoted by the empty diamond is from the two-photon transition between $|0\rangle$ and $|2\rangle$ states via four-wave mixing process. In this small $\beta/K$ regime, the KPO drive can be used to prepare the $|0\rangle$ and $|1\rangle$ states, as well as their superpositions. On the other hand, in the opposite limit $\beta/K \gg 1$, the oscillations are no longer sinusoidal (the rightmost plot in Fig. 1c). Instead, the dynamics within the time scale significantly shorter than the evolution driven by the Kerr ($\tau_{Rabi} \ll 1/K$) resembles those of a coherent state as indicated by the Wigner tomography in the right inset of Supplementary Fig. 3a.

The Rabi oscillation induced by the pump (Fig. 1d) can be understood similarly as described in Fig. 1e. The signal denoted by the filled triangle is induced by the transition between $|0\rangle$ and $|2\rangle$ states via three-wave mixing process, while the weak signal denoted by the empty triangle is from the transition between $|0\rangle$ and $|4\rangle$ states via six-wave mixing process. All data in Fig. 1c,d show excellent agreement with the simulation results, thus validating the use of Eq. (1) as the model for our KPO. (The simulation results can be found in Supplementary Fig. 3. For all simulations in this work, QuTiP was used[61,62]).

**Quantum interference and mapping from Fock to cat**
We generate cat states adiabatically using the pump pulse whose profile is $P\sin^2(\pi t/2\tau_{ramp})$, where the ramping time $\tau_{ramp}$ is 300 ns and $P/2\pi = 3.13$ MHz. At this pump amplitude, the KPO shows approximately −23 MHz of AC Stark shift in $2\omega_K$ (see Supplementary Fig. 3c for the measurement data). Thus, we change the pump frequency during

ramping, i.e., a chirped pump pulse, to compensate for the unwanted frequency shift without any DC pulses. Since $\tau_{ramp}$ is significantly less than $1/K$, where $K/2\pi = 3.1$ MHz after ramping up the pump, a counterdiabatic pulse is also used for faster mapping[63]. The profile of the counterdiabatic pulse is $0.3P\sin(\pi t/\tau_{ramp})$. The default experimental setting of pump detuning [$\Delta$ in Eq. (1)] is 1.0 MHz, except for the data in Figs. 2c and 4b.

Our key observation is an interference pattern in Wigner tomography, the definite signature of quantum coherence[64,65], of the even cat state $|+\mathrm{Cat}\rangle$ in the KPO. We also demonstrate one-to-one mapping from the cardinals in the Fock-basis Bloch sphere to those in the cat-basis Bloch sphere (Fig. 2a) by applying the pump to the $|1\rangle$ state as well as superpositions of $|0\rangle$ and $|1\rangle$. These results are shown in Fig. 2b. [Here, $|\pm i\mathrm{Cat}\rangle \equiv \mathcal{N}(|+\mathrm{Cat}\rangle \pm i|-\mathrm{Cat}\rangle)$ where $|-\mathrm{Cat}\rangle$ is the odd cat state and $\mathcal{N}$ is the normalization factor].

We find a clear coincidence between the theoretical position of classical energy minima and the measured size of cat states with various $\Delta$ values as shown in Fig. 2c. This finding provides concrete evidence that the interference pattern in Wigner tomography arises from the tunneling between two states confined by the Hamiltonian itself. [The classical energy can be obtained by replacing $\hat{a}$ and $\hat{a}^\dagger$ with complex numbers $\alpha$ and $\alpha^*$, respectively, in Eq. (1) with $\beta = 0$. The sizes of cat states are determined from the position where the measured Wigner function reaches its maximum after quantum interference is washed out].

To assess the fidelity of the mapping process from the Fock qubit space to the cat qubit space, we employ the following procedure. First, we obtain the density matrix of the KPO from the Wigner tomography using a neural-network algorithm called quantum state tomography with conditional generative adversarial network (QST-CGAN)[66,67], followed by the correction of unwanted Kerr evolution during the tomography process. Then, we construct the effective Fock and cat qubit density matrices from the full density matrix before and after the application of the pump, respectively. Using these sets of effective density matrices, we obtain the process fidelity by following the standard QPT procedure (see Fig. 6)[68]. The resulting process fidelity after the mapping is 0.757. The main sources of error are attributed to single-photon loss and fluctuations in the pump detuning, likely arising from imperfections in AC Stark shift cancellation (see Methods for details).

**Relaxation and dynamics of quantum interference**
As can be seen in Fig. 2c, the Wigner tomographies of cat states no longer exhibit the interference pattern after relaxation. A more systematic study on the relaxation process is presented in Fig. 3. We first generate the target cat state, wait for a specific time ($\tau_{relax}$ on the right side of Fig. 1b), and then perform Wigner tomography. From the tomography results, we calculate the populations of all six cardinals of the cat Bloch sphere.

The disappearance of the interference pattern is attributed to the reduction in the $|+\mathrm{Cat}\rangle$ population accompanied by an increase in the $|-\mathrm{Cat}\rangle$ population as shown in Fig. 3a; the resulting state after relaxation is a statistical mixture of two opposing cardinals. Since the transition from $|+\mathrm{Cat}\rangle$ to $|-\mathrm{Cat}\rangle$ changes the number parity, our result indicates that the primary relaxation mechanism is single-photon loss. However, other mechanisms, such as multiphoton loss and dephasing, cannot be neglected as indicated by the decreasing population of the qubit space (cross symbols in Fig. 3a,b).

Note that the populations of $|\pm\mathrm{Coh}\rangle$ and $|\pm i\mathrm{Cat}\rangle$ (the $xy$ plane of the Bloch sphere in Fig. 2a) oscillate with the frequencies $\delta f_x$ and $\delta f_y$, respectively. [Here, $|\pm\mathrm{Coh}\rangle \equiv \mathcal{N}(|+\mathrm{Cat}\rangle \pm |-\mathrm{Cat}\rangle)$]. The Wigner tomography results shown in Fig. 3b indicate that these states are nonstationary states, exhibiting tunneling back and forth from one classical energy minimum to the other (see Supplementary Movie 1). The resulting temporal oscillation in the quantum interference is a textbook example of dynamics associated with quantum tunneling[69].

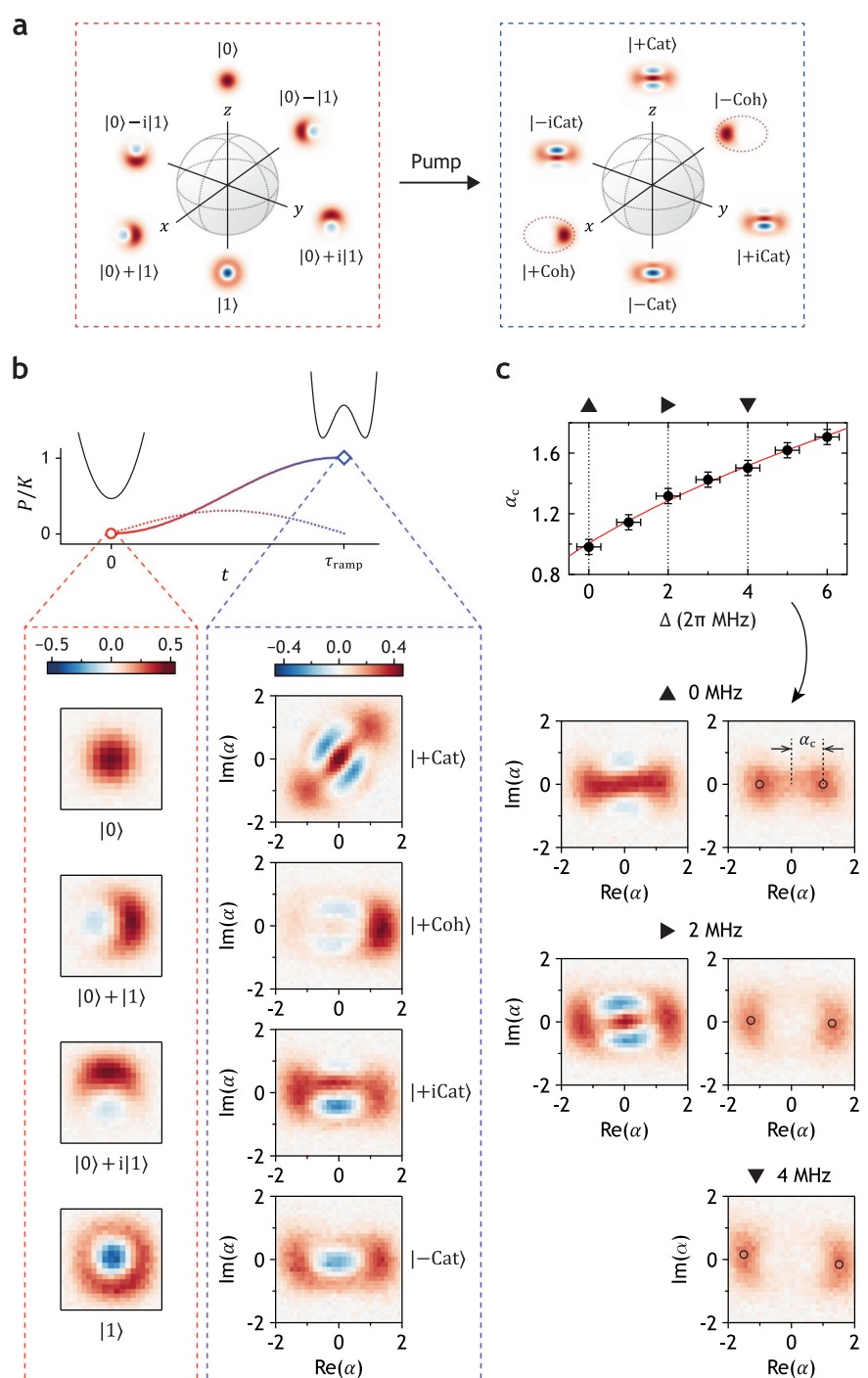

**Fig. 2 | Mapping from Fock to cat. a** Bloch sphere for the Fock state encoding (left) and the cat state encoding (right). **b** Experimental demonstration of one-to-one mapping from Fock states to cat states. The profile of the pump pulse is shown by the colored solid line; the dotted line shows the counterdiabatic pulse. The scales of Re($\alpha$) and Im($\alpha$) in Fock state tomography are both ±1.6. The Wigner tomography of the |+Cat⟩ state is intentionally rotated by adjusting the phase of the displacement pulse for an aesthetic reason. **c** Position of the classical energy minima $\alpha_c$ as a function of pump detuning [$\Delta$ in Eq. (1)]. The red solid line in the upper plot shows

the theoretical values following the formula $\alpha_c = \sqrt{(P + \Delta)/K}$[10]; the solid circles show the measured size of cat states. Error bars represent standard deviation; the errors in $\Delta$ are caused mainly by the slow drift of $\omega_K$ and frequent earthquakes in Japan. Wigner tomographies with three representative pump detunings are shown in the left part (before relaxation of quantum interference) and right part (after relaxation) of (c). The open black circles in the tomographies after relaxation indicate the classical energy minima.

The relaxation times and the frequency of this oscillation were extracted by fitting the population difference with $\exp(-\tau_{\text{relax}}/T_z)$ or $\cos(2\pi\delta f_{x(y)}\tau_{\text{relax}} + \phi_{x(y)})\exp(-\tau_{\text{relax}}/T_{x(y)})$ (black solid lines), where $\phi_{x(y)}$ is an offset phase. The fitting results are $T_z = 4.2\ \mu s$, $T_x = 6.6\ \mu s$, $T_y = 6.2\ \mu s$; $\delta f_x = 0.313$ MHz and $\delta f_y = 0.320$ MHz, resulting in an average value of 0.317 MHz. To understand the main factor that limits the

relaxation times of the cat states, we solve the Lindblad master equation with single-photon loss. We find that, to reproduce the measured relaxation times, the photon lifetime of the KPO must be approximately 10 $\mu s$, which closely matches the measured value of 8 $\mu s$. Thus, the relaxation times of the cat states are primarily constrained by the intrinsic photon lifetime of the KPO.

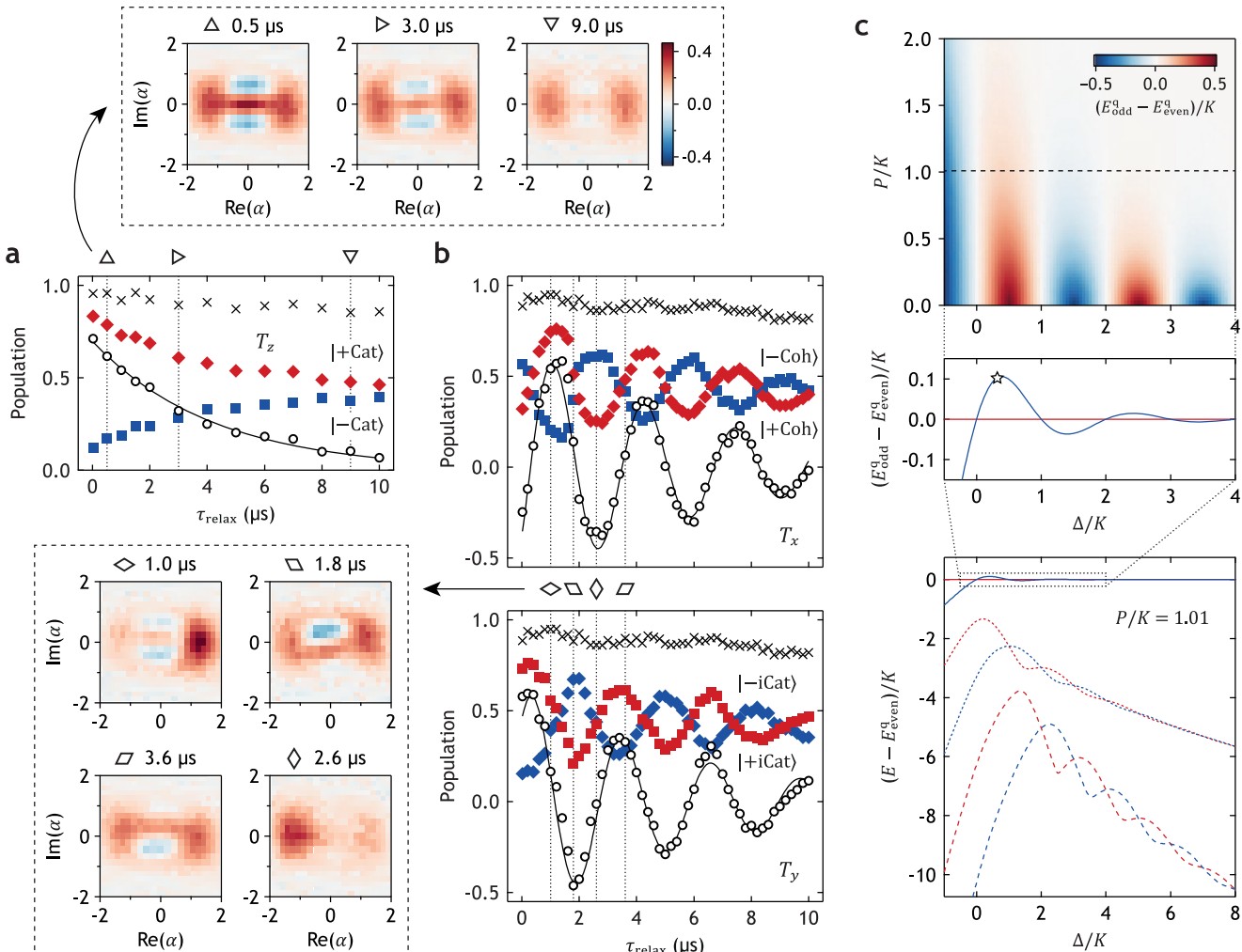

**Fig. 3 | Relaxation and temporal oscillation in quantum interference. a, b** Population of the cat qubit states along the *z* axis in Fig. 2a (a) and along the *x* and *y* axes (b). The cross and empty circle indicate the sum and difference between the populations of the states forming the qubit space, respectively. Black solid lines are fittings (see the main text). **c** The lower plot shows the first six quasienergy levels *E* calculated using Eq. (1) ($\beta = 0$), where $E^q_{even}$ is the energy of the even parity state in the qubit space. The red and blue levels indicate the number of parities of the states, even and odd, respectively. The quasienergy levels of the two lowest states are enlarged in the middle plot, where $E^q_{odd}$ is the energy of the odd parity state in the qubit space. The open star in the middle plot represents the average oscillation frequency extracted from (b). The color in the upper plot represents $(E^q_{odd} - E^q_{even})/K$. Our experimental condition $P/K = 1.01$ is denoted by the dashed line.

The extracted oscillation frequency indicates the difference in quasienergy between the two lowest states with different number parities, although their quasienergies appear as the highest values in the lower plot of Fig. 3c because of the minus sign in front of *K* in Eq. (1). (The quasienergies are eigenvalues of the system Hamiltonian in a rotating frame.) In the upper plot of Fig. 3c, the quasienergy difference decays exponentially along the *P* axis[53]. However, it oscillates sinusoidally with exponential decay along the Δ axis[30,54]. The calculated quasienergy difference is 0.318 MHz—an excellent agreement with the fitting result (the middle plot of Fig. 3c).

Given that the oscillation induced by the pump detuning corresponds to the clockwise rotation on the cat Bloch sphere, it acts as a background Z gate. This background Z gate makes the faster relaxation process dominant in the *xy* plane, resulting in almost identical $T_x$ and $T_y$ values, contrary to the results in refs. 52,53.

**Gate operation**
Lastly, we manipulate the quantum interference of the cat states by implementing gate operations. Since our computational basis is the even/odd cat state, X gate operation changes the number parity.

Therefore, the X gate is implemented by the drive term[11,12]. Unlike the typical X gate for a TLS, where the gate operation defines the reference phase, the KPO already has the phase reference—the pump. Thus, we need to sweep not only the X gate detuning $\Delta_d$ but also the phase relative to the pump $\phi_d$. The resulting Rabi oscillations in the parity of the KPO, which we call the cat Rabi, are shown in Fig. 4a. The $\phi_d$ dependence of the cat Rabi is consistent with the previous study[52]. The three Wigner tomographies confirm the rotation along the *x* axis, evolving from |+Cat⟩ to |−iCat⟩, and eventually reaching |−Cat⟩. From the simulation using Eq. (1), we extract $\beta/2\pi = 0.65$ MHz (see Supplementary Fig. 7a).

Note that the $\Delta_d$ dependence of the cat Rabi is qualitatively different from that of a typical TLS Rabi as shown in Fig. 1c,d. We find that a TLS with the transition frequency $\omega_{TLS}$ can reproduce such a Rabi pattern if we drive the TLS with two drive tones with opposite detuning, i.e., $\hat{\mathcal{H}}_{Rd}/\hbar = (\Omega_R/4)(e^{-i\Delta_{dT}t} + e^{+i\Delta_{dT}t})(\hat{\sigma}_+ + \hat{\sigma}_-)$ in the rotating frame, whereas the typical Rabi Hamiltonian for a TLS is $\hat{\mathcal{H}}_{Rs}/\hbar = (\Omega_R/2)(\hat{\sigma}_+ e^{+i\Delta_{dT}t} + \hat{\sigma}_- e^{-i\Delta_{dT}t})$. ($\Omega_R$ is the Rabi frequency, $\Delta_{dT} \equiv \omega_d - \omega_{TLS}$, and $\hat{\sigma}_\pm \equiv (\hat{\sigma}_x \pm i\hat{\sigma}_y)/2$, where $\hat{\sigma}_x$ and $\hat{\sigma}_y$ are Pauli operators. See Supplementary Fig. 7b for the simulations using these

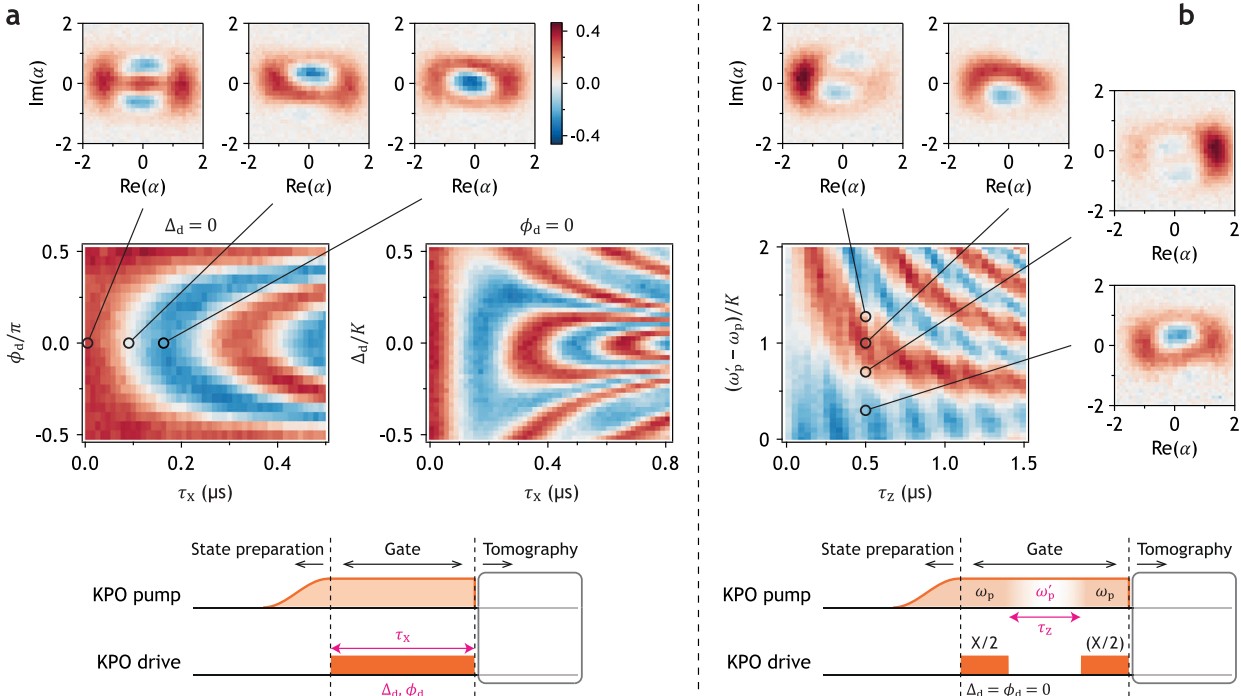

**Fig. 4 | Gate operation on the cat states.** The colors represent the value of the Wigner function at $\alpha = 0$, indicating the parity of the KPO. **a** Cat Rabi oscillations. Wigner tomographies at times corresponding to the 0, X/2, and X gates are shown above. **b** Cat Ramsey fringes. For the Wigner tomography, the second X/2 gate was not performed and is thus shown in parentheses. The pulse sequence for each measurement is shown at the bottom, with the pulses for the Wigner tomography are omitted for simplicity.

two Hamiltonians.) This suggests that when we map the dynamics of cat states in a KPO as that of a qubit, the drive detuning must be symmetrized. In other words, the combination of the Kerr nonlinearity and the pump not only generates an effective qubit space but also mixes the pump and the drive, generating the $\omega_p - \omega_d$ component similar to a typical microwave mixer. This component and the original drive frequency $\omega_d$ become $\pm |\omega_d - \omega_p/2|$ in the rotating frame. The $\phi_d$ dependence can also be reproduced by replacing $\Delta_{dT} t$ with $\phi_d$ in $\hat{\mathcal{H}}_{Rd}$.

A lesson from Fig. 3b is that the Z gate can be implemented by control of the temporal oscillation in the quantum interference, specifically the pump detuning. Thus, we implement the Z gate by increasing the pump frequency from $\omega_p$ to $\omega_p'$ and decreasing it back to $\omega_p$ (chirp), while the pump amplitude was kept constant. The profile of the frequency modulation is $(\omega_p' - \omega_p)\sin^2(\pi t/\tau_Z)$, where $\tau_Z$ is the Z gate time. By adding two X/2 gates before and after the Z gate, we observe Ramsey-like fringes in the parity of the KPO, which we call the cat Ramsey (Fig. 4b). The Wigner tomographies show that increasing $\omega_p'$ induces the counterclockwise rotation in the $xy$ plane of the Bloch sphere. (For the data in Fig. 4b, $\Delta/2\pi = 0.5$ MHz.)

Small background ripples in the cat Ramsey are an illustrative example of coherent errors resulting from imperfect X/2 gates. This imperfection arises due to $\beta$ not being sufficiently low compared to the energy gap between the cat states and the higher excitation levels. In our experimental conditions, the energy gap is approximately $1.4K(\approx 4.3$ MHz) (see Fig. 3c); therefore, the value of $\beta/2\pi$, which is 0.65 MHz, is small, albeit not negligibly smaller than the energy gap. Our simulation demonstrates that under such circumstances, the $\beta$ term in Eq. (1) behaves partly like a displacement operation, leading to population leakage. Subsequently, this error can be amplified under specific conditions of the Z gate (see Supplementary Fig. 7c–e). Since the energy gap increases roughly linearly with the size of the cat states, it is desirable to increase $P$ or $\Delta$ for faster and higher-quality X gate operation. In addition, careful selection of the Z gate condition is crucial to mitigate the effects of such imperfections.

We characterize our gate operation by QPT. The process fidelities after the X/2 and Z/2 gate operations are 0.844 and 0.794, respectively. The lower fidelity of the Z/2 gate is attributed to single-photon loss because the gate time for the Z/2 gate is 500 ns, whereas it is only 43 ns for the X/2 gate. (For details, see Methods.)

## Discussion

Rich physics and the potential usefulness of a KPO emerge from Hamiltonian engineering by the unique combination of moderate nonlinearity and the two-photon pump. However, a reliable characterization of the quantum state of a KPO has remained challenging because of its small nonlinearity for dispersive readout and simultaneously large nonlinearity for Wigner tomography.

In this work, we develop methods to resolve this problem by employing an ancillary two-level system and an advanced microwave pulse engineering technique. These enable us to directly observe the quantum interference of a cat state stabilized in a superconducting KPO through Wigner tomography. Moreover, we demonstrate one-to-one mapping between the Fock and cat cardinals. We establish the coincidence between the position of the classical energy minima and the size of the cat state, confirming that the observed quantum interference arises from tunneling between two bound states in phase space. We also investigate relaxation processes and dynamics of the quantum interference induced by the pump detuning, as well as the cat Rabi and Ramsey. Finally, we implement cat-state gate operations, demonstrating our control over the quantum coherence of the KPO.

This work experimentally reveals the essential quantum properties of a KPO, providing a solid foundation for future research on this system and also applicable to other driven quantum systems[31,70]. Furthermore, our KPO is a planar superconducting system, making it useful for practical applications. We believe that the unique quantum properties of a KPO bridge two leading paradigms in superconducting quantum information technology: TLS and QHO encoding.

## Methods

### Device

The chip for this work is shown in Fig. 5. The KPO (orange) is made of a series of asymmetric DC SQUIDs with a shunting capacitor. The reason for using asymmetric DC SQUIDs is to minimize the flux bias dependence of the Kerr coefficient while preserving resonance frequency tunability. The transmon (green) is capacitively coupled to the KPO. The readout resonator (purple) for the transmon is a quarter-wavelength resonator, which couples to another quarter-wavelength resonator to avoid the Purcell effect (Purcell filter). The devices on the right side are not used for this study. Since the measured frequency of the right KPO is about 140 MHz lower than that of the left KPO and the coupling strength is about 7 MHz, the contribution from the right KPO to the dynamics of the left KPO is negligible.

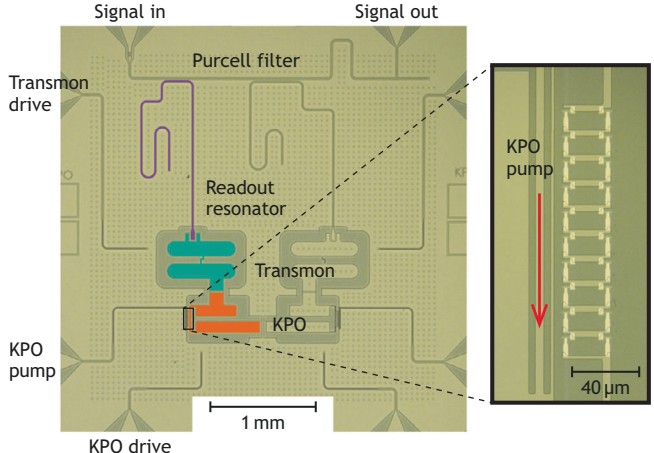

**Fig. 5 | Figure of the chip.** The figure on the right side shows the DC SQUIDs of the KPO.

The quarter-wavelength resonators and shunting capacitors were made of a 100 nm Nb film on a high-resistivity silicon substrate of 450 μm thickness. These devices were fabricated by maskless UV lithography and reactive-ion etching. Then, Josephson junctions were formed by the shadow evaporation technique using Dolan bridges. The thicknesses of the lower and upper Al films are 60 nm and 40 nm, respectively.

### Quantum process tomography

In this work, the mapping from the Fock to cat qubit spaces and the gate operations are characterized by the use of quantum process tomography (QST). The procedure for our QST is shown in Fig. 6. Although we closely followed the standard procedure[68], there are two differences: one is the use of QST-CGAN to obtain the density matrix of the KPO, and the other is the use of the effective qubit density matrix to treat the Fock and cat qubit spaces consistently. The effective qubit density matrices are defined as

$$\rho_F^q = \begin{pmatrix} \langle 0|\rho_F|0 \rangle & \langle 0|\rho_F|1 \rangle \\ \langle 1|\rho_F|0 \rangle & \langle 1|\rho_F|1 \rangle \end{pmatrix}, \quad (2)$$

$$\rho_C^q = \begin{pmatrix} \langle +\text{Cat}|\rho_C|+\text{Cat} \rangle & \langle +\text{Cat}|\rho_C|-\text{Cat} \rangle \\ \langle -\text{Cat}|\rho_C|+\text{Cat} \rangle & \langle -\text{Cat}|\rho_C|-\text{Cat} \rangle \end{pmatrix}, \quad (3)$$

where $\rho_F$ and $\rho_C$ are the full-density matrices from QST-CGAN with the Kerr correction before and after applying the pump, respectively. Here, the state basis $|\pm\text{Cat}\rangle$ was obtained by solving Eq. (1) with $\Delta/2\pi = 1.0$ MHz. Note that the effective qubit density matrices are not normalized so as not to discard population leakage from the qubit space. Then, we use the standard single-qubit operator basis:

$$\tilde{E}_0 = \frac{1}{2}\hat{I}, \tilde{E}_1 = \frac{1}{2}\hat{X}, \tilde{E}_2 = -\frac{i}{2}\hat{Y}, \tilde{E}_3 = \frac{1}{2}\hat{Z}. \quad (4)$$

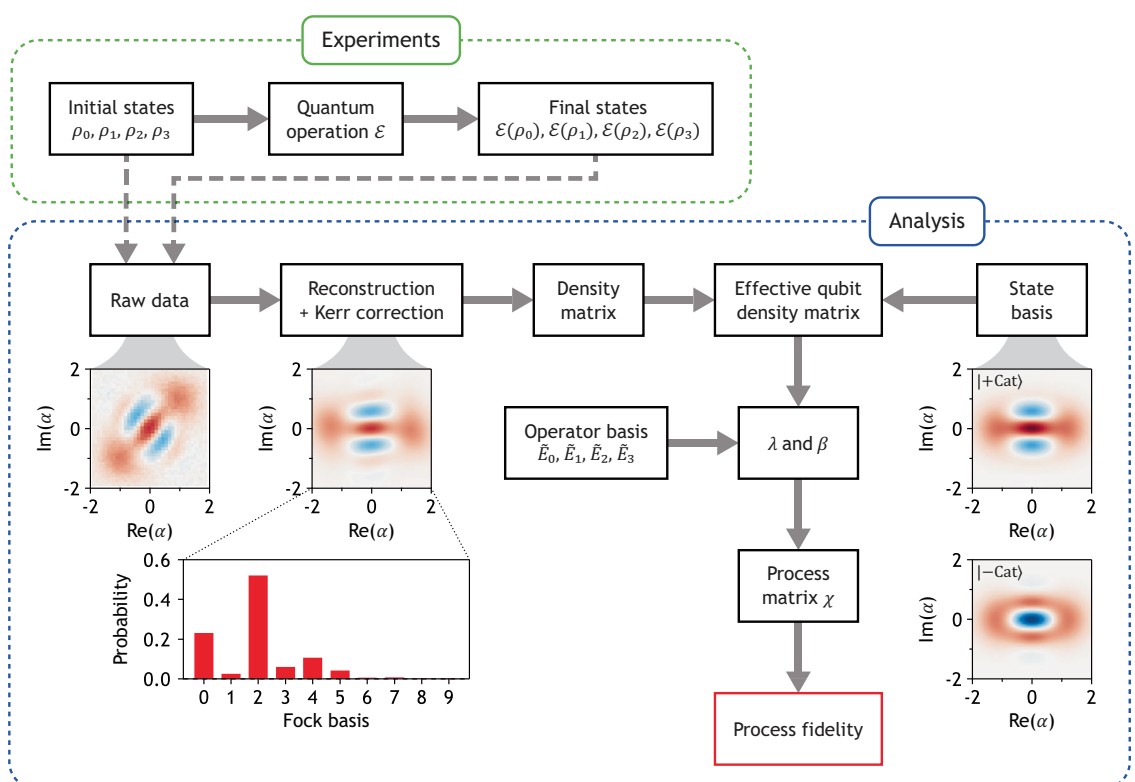

**Fig. 6 | Procedure for QPT.** Note that $\beta$ is not the drive amplitude here [See Eq. (5)].

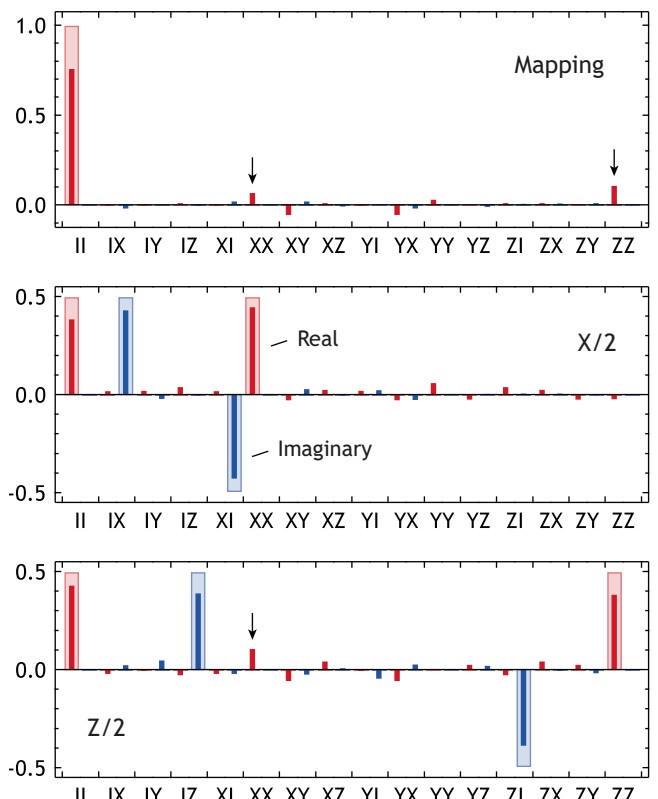

**Fig. 7 | Quantum process tomography.** Wide bars represent ideal mapping or gate operations, whereas narrow bars represent experimental results.

For completeness, we provide the definitions of matrices $\lambda$ and $\beta$ in Fig. 6. More detailed information can be found on p. 391 of ref. 68.

$$\mathcal{E}(\rho_j) = \sum_k \lambda_{jk}\rho_k, \quad \tilde{E}_m\rho_j\tilde{E}_n^\dagger = \sum_k \beta_{jk}^{mn}\rho_k. \quad (5)$$

The upper plot of Fig. 7 shows the process matrix obtained from $\rho_F^q$ (initial states) and $\rho_C^q$ (final states). The resulting process fidelity is 0.757. If the mapping is perfect, then $\rho_F^q$ and $\rho_C^q$ must be identical, i.e., the process must be fully described by the II component only. However, there are significant XX and ZZ components indicated by arrows. Since the X and Z operators represent the bit and phase flip channels, respectively[68], we conclude that the single-photon loss and fluctuation in the pump detuning are responsible for the mapping error.

We find that about 5% of the population leak out of the qubit space after the mapping (see the cross symbols in Fig. 3a, b). We believe that the dominant mechanism is imperfect AC Stark shift cancellation because a very rapid frequency modulation results in population leakage (see Sec. 6B in Supplementary Information). Thus, the AC Stark shift must be minimized at the stage of chip design.

For the gate operation QPT, all cat states were prepared from their counterparts in the Fock qubit space. This means that, once the pump was turned on, no gate operation was performed for state preparation to avoid any complications associated with gate operation on cat states such as the background ripples in Fig. 4b. Then, the process matrix was obtained from two sets of $\rho_C^q$—each set represents the states before and after the gate operation. The process fidelity after the X/2 and Z/2 gate operations is 0.844 and 0.794, respectively. The lower fidelity of the Z/2 gate can be attributed to single-photon loss as indicated by an arrow in the lower plot of Fig. 7. This is due to the longer gate time of the Z/2 gate, which is approximately one

order of magnitude longer than that of the X/2 gate (see Supplementary Table 2).

## Data availability
All data are available in the main text or the supplementary materials.

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

## Acknowledgements

We thank Tsuyoshi Yamamoto, Shiro Saito, Atsushi Noguchi, Kosuke Mizuno, Shotaro Shirai, Gopika Lakshmi Bhai, Hashizume Yoichiro,

and Fumiki Yoshihara for their interest in this project and helpful discussion. We also thank Kazumasa Makise in the National Astronomical Observatory of Japan for providing niobium films and the MIT Lincoln Laboratory for providing a Josephson traveling wave parametric amplifier. This work was supported by the Japan Science and Technology Agency (Moonshot R&D, JPMJMS2067; CREST, JPMJCR1676) and the New Energy and Industrial Technology Development Organization (NEDO, JPNP16007).

## Author contributions

S.K. and J.S.T. conceived the project. S.K., D.I., T.K., S.F., H.M., Y.Z., S.W., and J.S.T. designed the details of the experiment. D.I., T.K., S.F., and S.K. performed the measurements and data analysis. D.I., T.K., S.F., and S.K. performed the simulations with contributions from T.N. and J.J.X. S.W. provided theoretical support. D.I. and Y.Z. wrote the software for the measurements. H.M. and A.T. managed the hardware. S.K. and T.K. designed the chip. T.K. fabricated the chip with contributions from A.T. and R.W. S.K. wrote the original draft with contributions from S.F., T.K., and D.I. All authors contributed to the review and editing of the paper. S.K. and J.S.T. supervised the project. J.S.T. acquired the funding.

## Competing interests

The authors declare no competing interests.
