## [Peer review file · Nature Communications]

REVIEWER COMMENTS

Reviewer #1 (Remarks to the Author):

In "Observation and manipulation of quantum interference in a superconducting Kerr parametric oscillator" by Daisuke Iyama et al., the authors provide the first experimental Wigner tomography images of a quantum Kerr parametric oscillator (KPO). This is an important achievement and a significant technical feat. The manuscript covers all aspects of the system, from system characterization, to mapping between a two-level system and a KPO, to coherence measurements, and to gate operations.

The paper is a complex and demanding read. Being familiar with much of the terminology and phenomenology, I still needed to invest some time to understand the figures, which are packed with dense information. Nevertheless, I recommend the paper for publication without reservations. Not only does this paper provide a beautiful experimental demonstration of an important physical system, it is also scientifically rigorous and free from hyperbole. The graphs have been compiled with much attention to detail and with a didactic mindset, and will serve (apart from their obvious scientific value) as study material in many classrooms and journal clubs. The text is written as clearly as possible for such an advanced topic and contains much interesting information.

The only (!) comment I could come up with is that the time axis in the pulse sequences could be labelled as such to avoid any confusion.

Reviewer #2 (Remarks to the Author):

In this work, the authors demonstrate experimentally the possibility to create coherent superpositions of coherent states in a planar superconducting circuits architecture realizing a Kerr parametric oscillator (KPO). This, in contrast to state-of-the-art implementations which make use of microwave 3D cavities. In addition, the authors demonstrate direct Wigner tomography of the microwave field in a KPO by dispersively coupling the latter to an ancilla transmon instead of a linear readout resonator.

I find the paper well written and the results very relevant for the field of quantum computation with continuous variables. In particular, I appreciate the fact of moving back to planar geometries which becomes very important for scalability. For these reasons, I support the publication of this work after some minor comments are taken into account by the authors.

Comments:

In Eq. (1), the pump and drive amplitudes, P and β respectively, should be time-dependent functions in the most general case.

I don't understand what do the authors mean by the "resonance frequencies" indicated in Figs. 1 C and D with diamond markers. I don't understand their connection with the energy levels shown in Fig. 1E. I believe this really needs to be clarified in the manuscript.

In Fig. 1C on top of the contour plots there is this horizontal scale ranging from TLS to QHO. I understand QHO stands for "Quantum Harmonic Oscillator". Nevertheless, this is not mentioned anywhere on the main text. Also, would it be possible to make the label β / K bigger and/or to move it to the top of the scale? With so much information on this figure, this small label gets lost.

On page 2 of the main text the authors state in relation to Fig. 1C: "Instead, the short-time dynamics resemble those of a coherent state". What does this mean? Could the authors elaborate on the

dynamics a little bit more? In the QHO regime, the drive displaces the vacuum state creating a coherent state. Increasing the displacement reduces the vacuum state probability. Why at zero detuning, the authors observe revivals in the latter probability?

At the end of page 2, the authors state: "The sizes of cat states are determined from the position of red wavepackets in steady-state Wigner tomography". I find this a little bit confusing. I understand that by "red wavepackets" the authors refer to the displaced Gaussian functions which correspond to coherent states with opposite displacements. I would appreciate if the authors rephrased the above sentence a little bit better.

On page 3, section III the authors state: "The steady state of the KPO corresponds to a statistical mixture of two opposing cardinals, such as $|\pm \text{Cat}\rangle$, provides evidence of the energetic protection of cat states". The underlined sentence does not make any sense. Please check and remove/modify this.

On Fig. 2 the authors state "The Wigner tomography of the $|+\text{Cat}\rangle$ state is intentionally rotated by adjusting the phase of the displacement pulse." What does this mean? Why is this rotated? In the same figure the authors refer to "the Wigner probability of the steady-state". Can you define this?

Regarding the Wigner tomography section S5. I agree with the authors that for very short displacements one could approximate the displacement and the Kerr evolution as commuting operators. In this regime, the unitary (S4) should implement the correction. Maybe I'm wrong but I would expect that in the small-displacement-time regime, the following should hold $t_{\text{cor}} = \tau_d$. The authors observe a linear dependence though. Is it clear what the slope of this mean? Also, as τ_d is increased I would expect a departure from the linear relation. Maybe using the Baker-Campbell-Hausdorff relation the authors could elucidate corrections to (S4) as τ_d is increased.

On Fig. S9 the authors mention two parameters λ and β . I think that for the sake of completeness, these should be defined somewhere in the Supplementary Material.

I would like to point out very related works: <https://arxiv.org/abs/2210.09718> and <https://arxiv.org/abs/2308.14676>

In both, the authors study a SNAIL terminated superconducting resonator. The key feature of this device is that the Kerr non-linearity can be "turned on and off" via the flux modulation of the SNAIL. The authors use a dispersively coupled transmon qubit to characterize the non-linear resonator. In the second reference they used the transmon qubit to measure the Wigner function of the resonator field. Could you comment on these?

There is a typo in figure S5. It says "preperation". It should say "preparation".

Reviewer #3 (Remarks to the Author):

Kerr parametric oscillator Wigner tomography referee report

Referee: Jayameenakshi Venkatraman (jayavenkat@ucsb.edu: please note the change of address since the referee has moved affiliations.)

Manuscript Number: NCOMMS-23-34550-T

Author: Daisuke Iyama, Takahiko Kamiya, Shiori Fujii, Hiroto Mukai, Yu Zhou, Toshiaki Nagase, Akiyoshi Tomonaga, Rui Wang, Jiao-Jiao Xue, Shohei Watabe, Sangil Kwon, and Jaw-Shen Tsai

Title: Observation and manipulation of quantum interference in a superconducting Kerr parametric oscillator

1. Degree of originality:

Negligible [] Low [X] High []

2. Significance:

Not significant [] Marginal [X] Significant [] Highly significant []

3. Technical quality and scientific rigor:

3a) Is the reported work competently executed and technically correct?

Yes [X] Maybe [] No []

3b) Are the models or approximations used sufficiently justified?

Yes [X] Maybe [] No []

3c) Are the main conclusions or claims well supported or sufficiently substantiated?

Yes [] Maybe [] No [X]

Overall scientific quality: High [] Average [X] Low []

4. Quality of presentation:

Poor [] Marginal [] Average [] Good [X] Excellent []

II) Detailed assessment of the paper - intended for both the authors and the editors:

begin_report (use as much space as is needed). Please refrain from using gender-specific pronouns to refer to the author(s) (see Guidelines for Referees memo):

The main result of the current experimental work is the Wigner tomography of a Kerr parametric oscillator (KPO) implemented in a flux-modulated array of 10 asymmetric DC SQUIDs. Since the Wigner function is the expectation value of displaced parity, a measurement of the Wigner tomography needs the ability to make 1) strong displacements and 2) measure the parity of the KPO. Note that the measurement of displaced parity to measure the Wigner function is a well-known technique across various platforms such as trapped ions [1] and superconducting circuits [2]. The challenge in measuring the Wigner function for the KPO is that the quantum state that is being measured can get nontrivially distorted due to the noncommutativity between the Kerr nonlinearity and the displacement drive. However, a previous work [3] has overcome this challenge by performing Wigner tomography in both the transient and steady state by measuring the power spectral density of the emitted signal as it relaxes from an initial excited state to the ground state in a number split regime. In [3] however, since the KPO is strongly coupled to the environment to facilitate such readout, the coherence of the encoded state is quite poor. In the present work, the authors choose to protect the KPO by coupling it to a transmon instead which in turn is coupled to a readout resonator thereby shielding the KPO. However, both the longitudinal and transverse relaxation times of the KPO is <10us, which is quite disappointing. Note that recent works [4-6] have already achieved KPOs with much better coherences but weaker nonlinearity. Although, they have not shown Wigner tomography, they have presented both Ramsey and Rabi measurements which is equivalent to full tomography. This is all to say that while the present platform is of great interest to several groups, the work reported in the present article is incremental in originality compared to the existing literature. The authors present significant technological advancements such as tomography distortion correction with short and strong displacement pulses, counterdiabatic pulses for fast mapping, but this is too narrow in scope to justify publication in Nature Communications.

While the work is competently executed and technically correct, the referee has significant concerns about the claims as the referee elaborates next. Note that text from the paper is italicized in red.

MAJOR CONCERNS:

1. However, the state characterization method used in this work, transient power spectral density, requires the system to strongly couple to the environment. As a result, quantum coherence is significantly suppressed, making adiabatic cat state generation difficult [12, 18, 26, 33, 53, 54].

It appears that this argument is moot because in the present work the quantum coherence is significantly sacrificed anyway? The longitudinal and transverse relaxation time of the cat qubit is $3\mu\text{s}$ and $7\mu\text{s}$. What is limiting these coherences?

2. Although this work demonstrated mapping from Fock states to cat states and single-qubit gate operations, these demonstrations relied on support from simulations, as the dispersive readout does not provide full information on the quantum state of the KPO during the activation of the pump.

The author needs to clarify what the author means by support from simulations? Does the present work not derive support from simulations? These works which the author cites perform Rabi and Ramsey measurements. What are alternate explanations of the underlying measurements of tunneling and its periodic nulling due to interference presented in [6] that the authors can come up with?

3. The Pauli lifetimes have been measured for Δ / K different from an integer, even though the authors seem to understand that the tunneling in this case appears as a background Z gate that mixes the error channels. What motivates this choice? Furthermore, it would be nice to know the maximum noise bias the authors can achieve.

4. What sets the maximum cat size the authors can create? It looks like they cannot go beyond about a 1-2 photon cat. Is it nonlinear resonances in the transmon?

[1]: Leibfried, D. et al. Experimental determination of the motional quantum state of a trapped atom. *Phys. Rev. Lett.* 77, 4281–4285 (1996).

[2]: Hofheinz, M., Wang, H., Ansmann, M. et al. Synthesizing arbitrary quantum states in a superconducting resonator. *Nature* 459, 546–549 (2009)

[3]: Wang, Z., Pechal, M, Wollack E. A., et al. Quantum Dynamics of a Few-Photon Parametric Oscillator. *Physical Review X* 9, 021049 (2019)

[4]: Grimm, A., Frattini, N.E., Puri, S. et al. Stabilization and operation of a Kerr-cat qubit. *Nature* 584, 205–209 (2020)

[5]: The squeezed Kerr oscillator: spectral kissing and phase-flip robustness. Frattini, N. E., Cortiñas, R. G., Venkatraman J. et al. e-Print: 2209.03934 [quant-ph]

[6]: A driven quantum superconducting circuit with multiple tunable degeneracies. Venkatraman J., Cortiñas, R. G., Frattini, N. E. et al. e-Print: 2209.03934 [quant-ph]

We thank referees for valuable comments. In below, we address each comment one-by-one.

Reviewer #1 (Remarks to the Author):

In "Observation and manipulation of quantum interference in a superconducting Kerr parametric oscillator" by Daisuke Iyama et al., the authors provide the first experimental Wigner tomography images of a quantum Kerr parametric oscillator (KPO). This is an important achievement and a significant technical feat. The manuscript covers all aspects of the system, from system characterization, to mapping between a two-level system and a KPO, to coherence measurements, and to gate operations.

→ We appreciate spending your time for reviewing our manuscript and very positive response.

The paper is a complex and demanding read. Being familiar with much of the terminology and phenomenology, I still needed to invest some time to understand the figures, which are packed with dense information. Nevertheless, I recommend the paper for publication without reservations. Not only does this paper provide a beautiful experimental demonstration of an important physical system, it is also scientifically rigorous and free from hyperbole. The graphs have been compiled with much attention to detail and with a didactic mindset, and will serve (apart from their obvious scientific value) as study material in many classrooms and journal clubs. The text is written as clearly as possible for such an advanced topic and contains much interesting information.

→ We appreciate your recommendation of our paper to Nature Communications.

We are aware that the figures in this paper contain dense information. However, we have chosen to proceed in order to provide a solid and rigorous experimental foundation on this topic in a timely manner. We appreciate your understanding.

In the revised manuscript, we have included a movie clip constructed using the data from Fig. 3B to facilitate a better understanding of the dynamics of quantum interference. Additionally, we have cited E. Merzbacher's Physics Today article titled 'The Early History of Quantum Tunneling' (Ref. 68 in the revised manuscript) to establish a connection between our work and the long history of studies on quantum tunneling. We hope that these changes will enhance the reader's comprehension and overall experience with our paper.

For your convenience, we've included the paragraph containing Ref. 68 from the right column of page 3 in the revised manuscript. The modifications are highlighted:

Note that the populations of $|\pm\text{Coh}\rangle$ and $|\pm i\text{Cat}\rangle$ (the xy plane of the Bloch sphere in Fig. 2A) oscillate with the frequencies δf_x and δf_y , respectively. [Here, $|\pm\text{Coh}\rangle \equiv \mathcal{N}(|+\text{Cat}\rangle \pm |-\text{Cat}\rangle)$.] The Wigner tomography results shown in Fig. 3B indicate that these states are nonstationary states, exhibiting tunneling back and forth from one classical energy minimum to the other. **The resulting temporal oscillation in the quantum interference is a textbook example of dynamics associated with quantum tunneling [68].** The relaxation times and the frequency of this oscillation were extracted by fitting the population difference with $\exp(-\tau_{\text{relax}}/T_z)$ or $\cos(2\pi\delta f_{x(y)}\tau_{\text{relax}} + \phi_{x(y)})\exp(-\tau_{\text{relax}}/T_{x(y)})$ (black solid lines), where $\phi_{x(y)}$ is an offset phase. The fitting results are $T_z = 4.2 \mu\text{s}$, $T_x = 6.6 \mu\text{s}$, $T_y = 6.2 \mu\text{s}$; $\delta f_x = 0.313 \text{ MHz}$ and $\delta f_y = 0.320 \text{ MHz}$, resulting in an average value of 0.317 MHz . To understand the main factor that limits the relaxation times of the cat states, we solve the Lindblad master equation with single-photon loss. We find that, to reproduce the measured relaxation times, the photon lifetime of the KPO must be approximately $10 \mu\text{s}$, which closely matches the measured value of $8 \mu\text{s}$. Thus, the relaxation times of the cat states are primarily constrained by the intrinsic photon lifetime of the KPO.

The only (!) comment I could come up with is that the time axis in the pulse sequences could be labelled as such to avoid any confusion.

→ We've revised Figs. 1B and S6C. Below is a reproduction of the relevant part of Fig. 1 with the changes highlighted for your convenience:

We thank referees for valuable comments. In below, we address each comment one-by-one.

Reviewer #2 (Remarks to the Author):

In this work, the authors demonstrate experimentally the possibility to create coherent superpositions of coherent states in a planar superconducting circuits architecture realizing a Kerr parametric oscillator (KPO). This, in contrast to state-of-the-art implementations which make use of microwave 3D cavities. In addition, the authors demonstrate direct Wigner tomography of the microwave field in a KPO by dispersively coupling the latter to an ancilla transmon instead of a linear readout resonator.

I find the paper well written and the results very relevant for the field of quantum computation with continuous variables. In particular, I appreciate the fact of moving back to planar geometries which becomes very important for scalability. For these reasons, I support the publication of this work after some minor comments are taken into account by the authors.

→ We appreciate spending your time for reviewing our manuscript and very constructive comments.

Comments:

1. In Eq. (1), the pump and drive amplitudes, P and β respectively, should be time-dependent functions in the most general case.

→ Thank you for bringing this to our attention. We have rewritten Eq. (1), and you can find the updated equation below for your convenience:

$$\begin{aligned} \hat{\mathcal{H}}(t)/\hbar = & \Delta(t)\hat{a}^\dagger\hat{a} - \frac{K}{2}\hat{a}^\dagger\hat{a}^\dagger\hat{a}\hat{a} + \frac{P(t)}{2}(\hat{a}^\dagger\hat{a}^\dagger + \hat{a}\hat{a}) \\ & + \beta(t) \left[\hat{a}^\dagger e^{-i(\Delta_d t + \phi_d)} + \hat{a} e^{i(\Delta_d t + \phi_d)} \right]. \end{aligned} \quad (1)$$

2. I don't understand what do the authors mean by the "resonance frequencies" indicated in Figs. 1 C and D with diamond markers. I don't understand their connection with the energy levels shown in Fig. 1E. I believe this really needs to be clarified in the manuscript.

→ We apologize for this inconvenience. We have rewritten the relevant paragraphs, which are provided below (right column of page 2 in the revised manuscript) with the changes highlighted for your convenience.

The validity of this model is confirmed by measuring Rabi oscillations in the population of the KPO $|0\rangle$ state (Fig. 1C). The dynamics of the KPO are primarily governed by two parameters: β/K and P/K . In the limit of $\beta/K \ll 1$ (with $P/K = 0$ for simplicity), the dynamics of the KPO closely resemble those of a TLS (the leftmost plot in Fig. 1C). The Rabi oscillation denoted by the filled diamond is induced by the transition between $|0\rangle$ and $|1\rangle$ states described in Fig. 1E. The weak signal denoted by the empty diamond is from the two-photon transition between $|0\rangle$ and $|2\rangle$ states via four-wave mixing process. In this small β/K regime, the KPO drive can be used to prepare the $|0\rangle$ and $|1\rangle$ states, as well as their superpositions. On the other hand, in the opposite limit $\beta/K \gg 1$, the oscillations are no longer sinusoidal (the rightmost plot in Fig. 1C). Instead, the dynamics within the time scale significantly shorter than the evolution driven by the Kerr ($\tau_{\text{Rabi}} \ll 1/K$) resembles those of a coherent state as indicated by the Wigner tomography in the right inset of Fig. S4A.

The Rabi oscillation induced by the pump (Fig. 1D) can be understood similarly as described in Fig. 1E. The signal denoted by the filled triangle is induced by the transition between $|0\rangle$ and $|2\rangle$ states via three-wave mixing process, while the weak signal denoted by the empty triangle is from the transition between $|0\rangle$ and $|4\rangle$ states via six-wave mixing process. All data in Fig. 1C,D show excellent agreement with the simulation results, thus validating the use of Eq. (1) as the model for our KPO. (The simulation results can be found in Fig. S4. For all simulations in this work, QuTiP was used [61, 62].)

3. In Fig. 1C on top of the contour plots there is this horizontal scale ranging from TLS to QHO. I understand QHO stands for “Quantum Harmonic Oscillator”. Nevertheless, this is not mentioned anywhere on the main text. Also, would it be possible to make the label β/K bigger and/or to move it to the top of the scale? With so much information on this figure, this small label gets lost.

→ In our previous manuscript, we defined ‘QHO’ in the first paragraph of Sec. I on page 2. However, we understand that it may not be easily noticeable. When we prepare the initial manuscript, we made efforts to move this definition to the first page by rewriting the introduction but were unsuccessful. In this revised version, we have defined both ‘TLS’ and ‘QHO’ again in the caption of Fig. 1 for the convenience of readers. We have also enlarged the label β/K and explicitly expressed the TLS and QHO limits in terms of the β/K ratio. Below, we provide the relevant part of Fig. 1. The changes are highlighted.

FIG. 1. **System characterization.** (A) Rendered drawing of the chip and its circuit diagram. A cross symbol represents a Josephson junction. The KPO (orange) is composed of a series of 10 DC superconducting quantum interference devices (DC SQUIDS) with a shunting capacitor. This series of DC SQUIDS is indicated by two crosses with dots. The transmon (green) is capacitively coupled to the KPO. The readout resonator (purple) is a quarter-wavelength resonator. The system parameters can be found in Table S1. (B) Pulse sequence for Rabi oscillations and Wigner tomography. The control parameters for the Rabi oscillation measurements are shown in magenta. “ π ” and “ $\pi/2$ ” mean the transmon π - and $\pi/2$ -pulses, respectively. For the Wigner tomography, the first pulse in the KPO drive (empty orange pulse) is the Fock state preparation pulse. The Wigner tomography pulse sequence is used to obtain the data in Figs. 2 and 3. The pulse conditions are summarized in Table S2. (C,D) Rabi oscillations in the $|0\rangle$ state population of the KPO driven by the drive (C) and the pump (D). **TLS and QHO stand for two-level system and quantum harmonic oscillator, respectively.** (E) Energy level diagram of the KPO. The correspondence between the transitions and the observed signals is indicated by diamonds and triangles.

4. On page 2 of the main text the authors state in relation to Fig. 1C: “Instead, the short-time dynamics resemble those of a coherent state”. What does this mean? Could the authors elaborate on the dynamics a little bit more? In the QHO regime, the drive displaces the vacuum state creating a coherent state. Increasing the displacement reduces the vacuum state probability. Why at zero detuning, the authors observe revivals in the latter probability?

→ We acknowledge that we did not clarify the meaning of ‘short-time’ in our previous version. By ‘short-time’, we refer to a time scale significantly shorter than the evolution driven by the Kerr nonlinearity, $\tau_{Rabi} \ll 1/K$. If the time scale is comparable to or longer than the Kerr evolution, as you correctly pointed out, we observe revivals in the vacuum state probability. The revised section of the manuscript addressing this point is already included in your comment No. 2, which was located two pages prior.

5. At the end of page 2, the authors state: “The sizes of cat states are determined from the position of red wavepackets in steady-state Wigner tomography”. I find this a little bit confusing. I understand that by “red wavepackets” the authors refer to the displaced Gaussian functions which correspond to coherent states with opposite displacements. I would appreciate if the authors rephrased the above sentence a little bit better.

→ We have revised the sentence you referred to as follows (left column of page 3 in the revised manuscript):

We find a clear coincidence between the theoretical position of classical energy minima and the measured size of cat states with various Δ values as shown in Fig. 2C. This finding provides concrete evidence that the interference pattern in Wigner tomography arises from the tunneling between two states confined by the Hamiltonian itself. (The classical energy can be obtained by replacing \hat{a} and \hat{a}^\dagger with complex numbers α and α^* , respectively, in Eq. (1) with $\beta = 0$. The sizes of cat states are determined from the position where the measured Wigner function reaches its maximum after quantum interference is washed out.)

6. On page 3, section III the authors state: “The steady state of the KPO corresponds to a statistical mixture of two opposing cardinals, such as $|\pm \text{Cat}\rangle$, provides evidence of the energetic protection of cat states”. The underlined sentence does not make any sense. Please check and remove/modify this.

→ Thank you for bringing this to our attention. We have removed the sentence in question. The expression ‘a statistical mixture of two opposing cardinals’ has been slightly repositioned as shown below (left column of page 3 in the revised manuscript):

The disappearance of the interference pattern is attributed to the reduction in the $|+\text{Cat}\rangle$ population accompanied by an increase in the $|-\text{Cat}\rangle$ population as shown in Fig. 3A; the resulting state after relaxation is a statistical mixture of two opposing cardinals. Since the transition from $|+\text{Cat}\rangle$ to $|-\text{Cat}\rangle$ changes the number parity, our result indicates that the primary relaxation mechanism is single-photon loss. However, other mechanisms, such as multiphoton loss and dephasing, cannot be neglected as indicated by the decreasing population of

7. On Fig. 2 the authors state “The Wigner tomography of the $|+\text{Cat}\rangle$ state is intentionally rotated by adjusting the phase of the displacement pulse.” What does this mean? Why is this rotated?

→ In Wigner tomography, the coordinate corresponds to the amount of displacement in phase space. Therefore, the ‘angle’ of Wigner tomography is determined by the phase of the displacement pulse. We adjusted this phase for an aesthetic reason; we simply thought the tomography looked better this way. If it doesn’t appeal to your aesthetic sense, we apologize. We added “for an aesthetic reason” in the caption of Fig. 2. The figure is reproduced in the next page for your convenience.

8. In the same figure the authors refer to “the Wigner probability of the steady-state”. Can you define this?

→ In the revised manuscript, we have completely removed the term ‘steady-state’ because it is inaccurate due to the presence of multi-photon relaxation and dephasing processes. We have revised the sentence as shown below, highlighted for your convenience:

FIG. 2. Mapping from Fock to cat. (A) Bloch sphere for the Fock state encoding (left) and the cat state encoding (right). (B) Experimental demonstration of one-to-one mapping from Fock states to cat states. The profile of the pump pulse is shown by the colored solid line; the dotted line shows the counterdiabatic pulse. The scales of $\text{Re}(\alpha)$ and $\text{Im}(\alpha)$ in Fock state tomography are both ± 1.6 . The Wigner tomography of the $|+\text{Cat}\rangle$ state is intentionally rotated by adjusting the phase of the displacement pulse for an aesthetic reason. (C) Position of the classical energy minima α_c as a function of pump detuning [Δ in Eq. (1)]. The red solid line in the upper plot shows the theoretical values following the formula $\alpha_c = \sqrt{(P + \Delta)/K}$ [10]; the solid circles show the measured size of cat states. Wigner tomographies with three representative pump detunings are shown in the left part (before relaxation of quantum interference) and right part (after relaxation) of (C). The open black circles in the tomographies after relaxation indicate the classical energy minima.

9. Regarding the Wigner tomography section S5. I agree with the authors that for very short displacements one could approximate the displacement and the Kerr evolution as commuting operators. In this regime, the unitary (S4) should implement the correction. Maybe I'm wrong but I would expect that in the small-displacement-time regime, the following should hold $t_{cor} = \tau_d$. The authors observe a linear dependence though. Is it clear what the slope of this mean? Also, as τ_d is increased I would expect a departure from the linear relation. Maybe using the Baker-Cambpell-Hausdorff relation the authors could elucidate corrections to (S4) as τ_d is increased.

→ The crucial thing is that the ratio t_{cor}/τ_d depends on the “shape” of the displacement pulse. Notably, the Kerr nonlinear term commutes with the parity operator, meaning that pure Kerr evolution (when the displacement pulse amplitude is zero) does not alter the parity of the system. What influences or distorts the parity measurement is the noncommutativity between the Kerr term and the single-photon drive term. Consequently, the error in the parity measurement depends on the amplitude of the single-photon drive at each instance. Thus, $t_{cor}/\tau_d \neq 1$. Unfortunately, we couldn't derive a reliable mathematical expression for t_{cor}/τ_d .

10. On Fig. S9 the authors mention two parameters λ and β . I think that for the sake of completeness, these should be defined somewhere in the Supplementary Material.

→ We appreciate your suggestion. These matrices are defined on the last page of the Supplementary Material. For your convenience, we have reproduced them here:

For completeness, we provide the definitions of matrices λ and β in Fig. S9. More detailed information can be found on p. 391 of Ref. [69].

$$\mathcal{E}(\rho_j) = \sum_k \lambda_{jk} \rho_k, \quad \tilde{E}_m \rho_j \tilde{E}_n^\dagger = \sum_k \beta_{jk}^{mn} \rho_k.$$

11. I would like to point out very related works: <https://arxiv.org/abs/2210.09718> and <https://arxiv.org/abs/2308.14676>

In both, the authors study a SNAIL terminated superconducting resonator. The key feature of this device is that the Kerr non-linearity can be “turned on and off” via the flux modulation of the SNAIL. The authors use a dispersively coupled transmon qubit to characterize the non-linear resonator. In the second reference they used the transmon qubit to measure the Wigner function of the resonator field. Could you comment on these?

→ We are pleased to have come across these related works, which strengthen our approach to handling bosonic code in a planar superconducting system. By activating the Kerr nonlinearity, we can rapidly create a cat state without the need for additional gate operations. One notable advantage is that, when the Kerr nonlinearity is turned off, this approach remains compatible with most well-established techniques for bosonic codes. From our perspective, these two works are advancing the conventional bosonic code paradigm by introducing an additional control parameter (Kerr nonlinearity) and returning to a planar geometry, which offers greater scalability potential. We have cited these works as Refs. 36 and 59 in the revised manuscript. Our work aims to chart a novel path that bridges the gap between two paradigms: two-level system encoding and harmonic oscillator encoding. It is very interesting to see that the Kerr nonlinearity plays the key ingredient for all these approaches.

12. There is a typo in figure S5. It says “preperation”. It should say “preparation”.

→ Thanks for pointing out the typo. We corrected it in the revised manuscript.

We thank referees for valuable comments. In below, we address each comment one-by-one.

Reviewer #3 (Remarks to the Author):

The main result of the current experimental work is the Wigner tomography of a Kerr parametric oscillator (KPO) implemented in a flux-modulated array of 10 asymmetric DC SQUIDs. Since the Wigner function is the expectation value of displaced parity, a measurement of the Wigner tomography needs the ability to make 1) strong displacements and 2) measure the parity of the KPO. Note that the measurement of displaced parity to measure the Wigner function is a well-known technique across various platforms such as trapped ions [1] and superconducting circuits [2]. The challenge in measuring the Wigner function for the KPO is that the quantum state that is being measured can get nontrivially distorted due to the noncommutativity between the Kerr nonlinearity and the displacement drive. However, a previous work [3] has overcome this challenge by performing Wigner tomography in both the transient and steady state by measuring the power spectral density of the emitted signal as it relaxes from an initial excited state to the ground state in a number split regime. In [3] however, since the KPO is strongly coupled to the environment to facilitate such readout, the coherence of the encoded state is quite poor. In the present work, the authors choose to protect the KPO by coupling it to a transmon instead which in turn is coupled to a readout resonator thereby shielding the KPO. However, both the longitudinal and transverse relaxation times of the KPO is $<10\mu\text{s}$, which is quite disappointing. Note that recent works [4-6] have already achieved KPOs with much better coherences but weaker nonlinearity. Although, they have not shown Wigner tomography, they have presented both Ramsey and Rabi measurements which is equivalent to full tomography.

→ First and foremost, we appreciate the time you've devoted to reviewing our manuscript.

It's important to note that Rabi oscillation should not be equated with state tomography, even in the case of an ideal two-level system (TLS). Rabi oscillation represents oscillations in the populations of the $|0\rangle$ and $|1\rangle$ states, which correspond to the diagonal elements of the density matrix. However, it does not provide information about the off-diagonal elements of the density matrix. This is precisely why quantum state tomography is crucial. The complexity is further amplified in the context of bosonic codes, as even the diagonal elements of the density matrix cannot be fully derived from Rabi measurements in a multi-level system. While it might be argued that the cat states in a Kerr parametric oscillator (KPO) benefit from an energy gap, it's essential to recognize that the magnitude of this energy gap is several orders of magnitude smaller than the anharmonicity observed in conventional qubits like transmons or flux qubits. Even in these conventional qubits, population leakage is a known consideration. Therefore, in a system with a limited protection gap, it becomes imperative to address non-idealities in cat states, including population leakage and slight distortions in Wigner tomography.

Note that our work is not primarily focused on achieving the longest coherence time or addressing biased noise. Hence, comparing our coherence time with other studies may not be directly relevant to the goals and objectives of our research.

This is all to say that while the present platform is of great interest to several groups, the work reported in the present article is incremental in originality compared to the existing literature. The authors present significant technological advancements such as tomography distortion correction with short and strong displacement pulses, counterdiabatic pulses for fast mapping, but this is too narrow in scope to justify publication in Nature Communications.

→ While distortion correction in tomography is certainly a notable technical advancement in our work, we emphasize that it's not the primary focus. The central achievement of our study lies in providing a rigorous and timely elucidation of the fundamental quantum properties of a KPO.

While the work is competently executed and technically correct, the referee has significant concerns about the claims as the referee elaborates next. Note that text from the paper is italicized in red.

MAJOR CONCERNS:

1. *"However, the state characterization method used in this work, transient power spectral density, requires the system to strongly couple to the environment. As a result, quantum coherence is significantly suppressed, making adiabatic cat state generation difficult [12, 18, 26, 33, 53, 54]."*

It appears that this argument is moot because in the present work the quantum coherence is significantly sacrificed anyway? The longitudinal and transverse relaxation time of the cat qubit is 3 μ s and 7 μ s. What is limiting these coherences?

→ It's important to note that the terms 'long' and 'short' are relative, particularly in the realm of science. In the quotation you referenced, we explicitly defined the standard for quantum coherence as the timescale required for adiabatic cat state generation. To ensure adiabaticity, the cat creation time must be approximately 20 times longer than the inverse of the Kerr coefficient, which, for our system, amounts to about 50 ns. Consequently, the pulse length for high-fidelity cat creation should be around 1 μs , and we were able to reduce it to 0.3 μs using a counterdiabatic pulse. In our work, the photon lifetime (8 μs) significantly exceeds the cat state generation time by more than an order of magnitude. From this perspective, the coherence time of our system is sufficiently long. In contrast, in Ref. 51 (Stanford work) of the revised manuscript, the photon lifetime is more than an order of magnitude shorter (150 ns) than ours, and the pulse length required for high-fidelity cat generation should be at least about 180 ns (approximately 20/K), which surpasses the photon lifetime. In Ref. 51, the majority of photon loss is caused by strong coupling to the environment. Therefore, our argument remains valid.

Regarding your question, our simulations demonstrate that to achieve relaxation times of 4 μs along the z-axis and 6 μs in the xy-plane, the photon lifetime must be approximately 10 μs , closely aligning with the measured value of 8 μs . Thus, the relaxation times of our cat states are primarily constrained by the intrinsic photon lifetime. Additionally, the small P/K ratio and mixed error channel due to finite detuning limit the noise bias. For your convenience, we have included the revised paragraph here (right column of page 3 in the revised manuscript):

Note that the populations of $|\pm\text{Coh}\rangle$ and $|\pm i\text{Cat}\rangle$ (the xy plane of the Bloch sphere in Fig. 2A) oscillate with the frequencies δf_x and δf_y , respectively. [Here, $|\pm\text{Coh}\rangle \equiv \mathcal{N}(|+\text{Cat}\rangle \pm |-\text{Cat}\rangle)$.] The Wigner tomography results shown in Fig. 3B indicate that these states are nonstationary states, exhibiting tunneling back and forth from one classical energy minimum to the other. The resulting temporal oscillation in the quantum interference is a textbook example of dynamics associated with quantum tunneling [68]. The relaxation times and the frequency of this oscillation were extracted by fitting the population difference with $\exp(-\tau_{\text{relax}}/T_z)$ or $\cos(2\pi\delta f_{x(y)}\tau_{\text{relax}} + \phi_{x(y)})\exp(-\tau_{\text{relax}}/T_{x(y)})$ (black solid lines), where $\phi_{x(y)}$ is an offset phase. The fitting results are $T_z = 4.2 \mu\text{s}$, $T_x = 6.6 \mu\text{s}$, $T_y = 6.2 \mu\text{s}$; $\delta f_x = 0.313 \text{ MHz}$ and $\delta f_y = 0.320 \text{ MHz}$, resulting in an average value of 0.317 MHz. To understand the main factor that limits the relaxation times of the cat states, we solve the Lindblad master equation with single-photon loss. We find that, to reproduce the measured relaxation times, the photon lifetime of the KPO must be approximately 10 μs , which closely matches the measured value of 8 μs . Thus, the relaxation times of the cat states are primarily constrained by the intrinsic photon lifetime of the KPO.

2. *“Although this work demonstrated mapping from Fock states to cat states and single-qubit gate operations, these demonstrations relied on support from simulations, as the dispersive readout does not provide full information on the quantum state of the KPO during the activation of the pump.”*

The author needs to clarify what the author means by support from simulations? Does the present work not derive support from simulations? These works which the author cites perform Rabi and Ramsey measurements. What are alternate explanations of the underlying measurements of tunneling and its periodic nulling due to interference presented in [6] that the authors can come up with?

→ First and foremost, we emphasize that our work complements and strengthens the contributions of the Yale group; our intention is not to overly critique their work or propose alternative explanations.

In principle, experimental demonstrations of mapping between two quantum states are considered incomplete without full state characterization. In our study, we conducted comprehensive state characterizations both before and after applying the pump, thus showcasing the mapping without any reliance on simulations or prior assumptions.

In Ref. 52 of the revised manuscript, the quantum states with and without the pump were not fully characterized. The primary basis for creating a cat state in this work is that the observed Rabi oscillations align with simulations. (Note that, as mentioned earlier, Rabi oscillation does not equate to full state characterization.) While Refs. 53 and 54 performed meticulous spectroscopy measurements and extensively explored relaxation times, these works do not offer direct observational evidence of the mapping from Fock to cat states. In these studies, having cat states serves as an ‘explanation’ rather than an ‘observation’.

3. *The Pauli lifetimes have been measured for Δ / K different from an integer, even though the authors seem to understand that the tunneling in this case appears as a background Z gate that mixes the error channels. What motivates this choice? Furthermore, it would be nice to know the maximum noise bias the authors can achieve.*

→ Our primary focus was to detect the ‘dynamics’ of quantum interference and demonstrate that its origin lies in phase space tunneling. For this reason, we intentionally selected a non-integer Δ/K ratio. As mentioned earlier, our primary concern did not involve biased noise. Given our constrained P/K ratio, the maximum noise bias in our work will be significantly lower than that observed in the Yale group's studies.

4. *What sets the maximum cat size the authors can create? It looks like they cannot go beyond about a 1-2 photon cat. Is it nonlinear resonances in the transmon?*

→ Photon number splitting data (Fig. S4D) demonstrates that our KPO can have a mean photon number significantly greater than 2. This implies that neither the transmon nor the single-photon drive on the KPO constrains the cat's size. As previously illustrated in Fig. S4C, the size of our cat states is limited by the divergence of the Kerr coefficient, the mechanism of which remains unknown at this stage.

List of changes

1. We include a movie clip that shows the dynamics of quantum interference (Comment from Reviewer 1).
2. E. Merzbacher's Physics Today article titled 'The Early History of Quantum Tunneling' (Ref. 68 in the revised manuscript) has been cited (Comment from Reviewer 1).
3. We added the label 'Time' in Fig. 1B (Comment from Reviewer 1).
4. Equation (1) shows time-dependent functions explicitly (Comment 1 from Reviewer 2).
5. The paragraph starting with "The validity of this model..." was revised (right column of page 2 in the revised manuscript) (Comments 2 and 4 from Reviewer 2).
6. The definitions of TLS and QHO were included in the caption of Fig. 1 (Comment 3 from Reviewer 2).
7. The paragraph starting with "We find a clear coincidence between..." was revised (left column of page 3 in the revised manuscript) (Comment 5 from Reviewer 2).
8. The paragraph starting with "The disappearance of the interference pattern ..." was revised (left column of page 3 in the revised manuscript) (Comment 6 from Reviewer 2).
9. The term 'steady-state' was completely removed, and the caption of Fig. 2 was revised accordingly (Comment 8 from Reviewer 2).
10. The definition of matrices λ and β are now provided on the last page of the Supplementary Material (Comment 10 from Reviewer 2).
11. References 36 and 59 were added (Comment 11 from Reviewer 2).
12. A typo in Fig. S5 was corrected (Comment 12 from Reviewer 2).
13. The paragraph starting with "Note that the populations of..." was revised (right column of page 3 in the revised manuscript) (Comment 1 from Reviewer 3).
14. The title of Sec. III was rewritten.
15. Reference 57 was added.

REVIEWERS' COMMENTS

Reviewer #2 (Remarks to the Author):

The authors have addressed all of the comments in my original report. Therefore, I support the publication of this work.

Reviewer #3 (Remarks to the Author):

Satisfied with the changes